# Estimating a novel stochastic model for within-field disease dynamics of banana bunchy top virus via approximate Bayesian computation

Abhishek Varghese[1,2]*, Christopher Drovandi[1,2], Antonietta Mira[3,4], Kerrie Mengersen[1,2]

**1** School of Mathematical Sciences, Queensland University of Technology, Brisbane, Australia, **2** ARC Centre for Excellence in Mathematical and Statistical Frontiers (ACEMS), Brisbane, Australia, **3** Institute of Computational Science, Università della Svizzera italiana, Lugano, Switzerland, **4** Department of Science and High Technology, Università degli Studi dell'Insubria, Como, Italy

\* abhishek.varghese@connect.qut.edu.au

**Data Availability Statement:** All relevant data and code are found within the manuscript and its Supporting Information files. They are also made

## Abstract

The Banana Bunchy Top Virus (BBTV) is one of the most economically important vector-borne banana diseases throughout the Asia-Pacific Basin and presents a significant challenge to the agricultural sector. Current models of BBTV are largely deterministic, limited by an incomplete understanding of interactions in complex natural systems, and the appropriate identification of parameters. A stochastic network-based Susceptible-Infected-Susceptible model has been created which simulates the spread of BBTV across the subsections of a banana plantation, parameterising nodal recovery, neighbouring and distant infectivity across summer and winter. Findings from posterior results achieved through Markov Chain Monte Carlo approach to approximate Bayesian computation suggest seasonality in all parameters, which are influenced by correlated changes in inspection accuracy, temperatures and aphid activity. This paper demonstrates how the model may be used for monitoring and forecasting of various disease management strategies to support policy-level decision making.

## Author summary

The Banana Bunchy Top Virus (BBTV) poses one of the greatest threats to the food security of developing nations and the banana industry throughout the Asia-Pacific Basin. Decision-makers face significant challenges in mitigating BBTV spread in banana plantations due to the vector-borne spread of this disease, which is significantly influenced by a vast array of external environmental factors that are unique to each plantation. We propose a flexible network-based model that describes the spread of BBTV in a real banana plantation through a random process while accounting for individual plantation characteristics and utilise a principled methodology for estimating model parameters. Our models can be used to quantify the effects of seasonal changes and plantation configuration on

available at the following Github repository: https://github.com/Sheksta/bbtv-abc

**Funding:** A.V received funding by the Australian Research Council (ARC) Centre of Excellence for Mathematical and Statistical Frontiers (ACEMS) and the Queensland University of Technology under grant number CE140100049. K.M was supported by an ARC Laureate Fellowship under grant number FL150100150. C.D was supported by the ARC Discovery Project and the Queensland University of Technology under grant number DP200102101. A.M's role in this paper was self-funded. The funders had no role in study design, data collection and analysis, decision to publish, or preparation of the manuscript.

**Competing interests:** The authors have declared that no competing interests exist.

BBTV spread and can be used to predict high-risk areas in this plantation. We believe that our model might be used by decision-makers to evaluate the effectiveness of current disease management strategies and explore opportunities for improvements.

## Introduction

The Banana Bunchy Top Virus (BBTV) is one of the most economically important vector-borne banana diseases throughout the Asia-Pacific Basin. The disease was first introduced to Australia in 1913 via infected suckers from Fiji, and spread locally through the banana aphid, *Pentalonia nigronervosa* [1]. With limited knowledge on epidemiological characteristics of the disease or disease management approaches, incidence rates across Australian banana plantations rose rapidly, eradicating over 90% of national crop production in the 1930's [2]. Cook et al. [3] estimate that the economic benefits of BBTV exclusion from commercial plantations range from $15.9 to $20.7 million each year, approximately 5% of annual crop production value. Aggressive disease management strategies implemented by the Australian Government from the 1930s-90s have largely restricted the disease to the South-East Queensland and Northern New South Wales regions of Australia [4]. Eradication, however, has not been achieved, requiring continuous monitoring by the National BBTV Program.

While monitoring the infection counts across the region does provide an indication of disease management success, a vast array of external environmental factors influence BBTV growth, making this an unreliable metric. Furthermore, there are currently limited opportunities to explore various management strategies for BBTV within banana plantations, which could reduce infection rates and identify cost-saving measures for the monitoring program. In such scenarios, mathematical models offer the opportunity to simulate various disease management strategies with a low-cost and quick turnaround [5].

Unfortunately, there have been few contributions to modelling the disease dynamics of BBTV in plantations in the last few decades–despite the significant advancements in computational resources and our understanding of vector-borne diseases. Allen [6] generated a stochastic spatiotemporal polycyclic model for BBTV to describe disease progress within a banana plantation, specifically focusing on identifying the mean inoculation distance of BBTV. However, the model was designed for a hypothetical homogenous circular plantation, which does not account for the various plantation configurations and unique plantation characteristics present around the world; a key factor which greatly affects the effectiveness of disease management strategies. Another model developed by Smith et al. [1] aimed to describe the influence of external inoculum on BBTV spread within a banana plantation in the Philippines. However, the model developed by Smith et al. [1] is deterministic, which results in poor representations of complex natural processes that are inherently probabilistic. Furthermore, these articles do not provide a principled parameter estimation method with appropriate uncertainty quantification based on available field data.

In our paper, we propose a new stochastic model that describes BBTV spread across a banana plantation, parameterising for neighbouring and long-distance infectivity rates, and recovery rates. Further, we develop a principled Bayesian parameter estimation method for calibrating this model to real field data. Given the intractability of the likelihood function for this model, we employ approximate Bayesian computation (ABC) [7] for estimating model parameters and their uncertainty. Our methodology is inspired by Dutta et al. [8], who demonstrate that ABC may be effectively used to estimate the spreading parameters of a disease by applying a simple Susceptible-Infected-Susceptible (SIS) model over a known network

structure. This paper adapts and extends this approach to further understand the spreading characteristics of BBTV, evaluates various disease management strategies at the plantation level, and predicts the spread of future outbreaks. Even though our work is motivated by the vector-borne transmission of BBTV, we believe that our modelling framework is easily adaptable to describe the within-field disease dynamics of other vector-borne diseases.

## Methods

This study focuses on a banana plantation in Newrybar, near the North-Eastern border of New South Wales in Australia (28˚42'14.8"S, 153˚32'20.4"E), with an area of approximately 12 hectares. A routine site inspection in 2013 identified a banana plant infected with BBTV in the North-Western region of the farm. A following inspection in 2014 identified 26 infections clustered across the South-Eastern region of the farm. Since 2014, the site has undergone monthly inspections, collecting location data and plant characteristics, while implementing a rogue-and-remove disease management strategy. The location of every infected plant has been recorded using the Global Positioning System (GPS) functionality on a smartphone. The data-set consists of 38 snapshots of infection data at monthly intervals with the coordinates of each infected plant identified and rogued during a site visit to the farm.

Fig 1 provides a birds-eye view of the plantation. The plantation is separated by dirt paths approximately 3–5 m wide, creating smaller subsections of banana plants across the plantation.

Fig 2(A) provides the BBTV infection counts of the Newrybar site at monthly intervals from December 2014 to January 2018. Since December 2014, over 3000 banana plants have been removed from the Newrybar plantation.

Fig 2(B) highlights the seasonality in BBTV infectivity. Over the observation period of approximately 38 months, BBTV infection counts tend to peak during warmer periods of the year (November to February), while receding during traditionally colder months (May to August).

### BBTV forward-simulation model

We propose to model the spread of BBTV in a banana plantation by modifying the 'simple contagion' model developed by Dutta et al. [8]. The 'simple contagion' model simulates a standard SIS process on a fixed network structure. At every time step, each infected node chooses one of

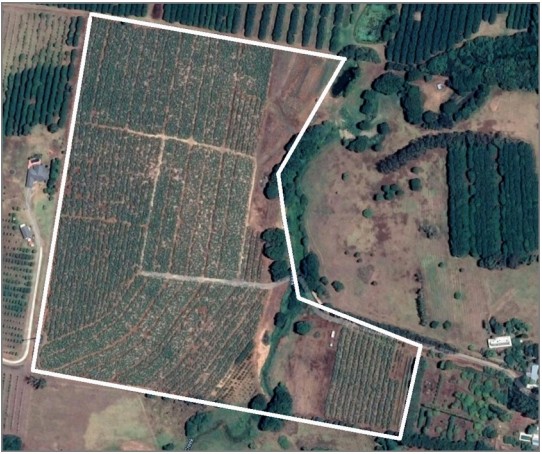

**Fig 1. Satellite image of Newrybar banana plantation.** Solid white lines indicate the approximate border of the property.

(a)

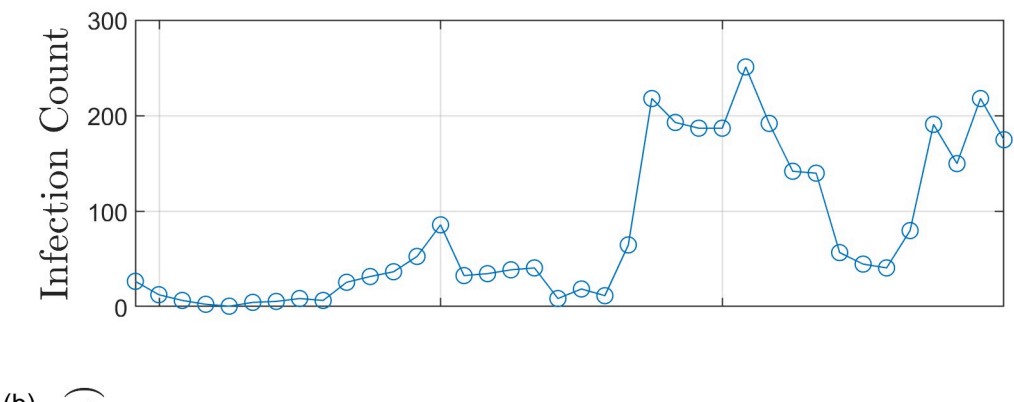

(b)

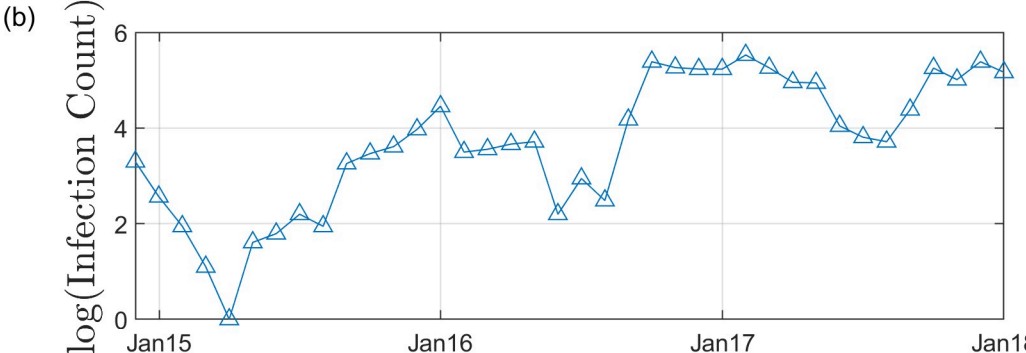

**Fig 2. BBTV infection counts since surveying began in Dec-2014.** (a) BBTV Infection counts over the survey time period (38 months). Surveys occurred at approximately monthly intervals (between 25–30 days), according to the National Banana Bunchy Top Virus Project disease management practices. Therefore, the exact date of plantation survey has been disregarded. (b) Logarithmic transformation of BBTV infection counts, to highlight seasonal variance in infectivity.

its neighbours with equal probability regardless of their status (susceptible or infected), and if the chosen node is susceptible, it is infected with probability θ. Our network-based forward-simulating modelling approach enables easy adaptation to describe the disease dynamics over a range of plantations, and various vector-borne diseases. A network representation of the banana plantation and its degree distribution are provided in Fig 3(A) and 3(B), respectively.

Dutta et al. [8] denote the 'simple contagion' model by $M_s$ and parameterise it in terms of the spreading rate θ and the seed node $n_{sn}$. For given values of these two parameters, they forward simulate the evolving epidemic over time using model $M_s$.

We adapt and extend model $M_s$ in several ways for our application. Dutta et al. [8] determine a node to be an individual person, with edges representing each person's contacts. The Newrybar banana plantation has a plant density of approximately 4000 banana plants per hectare, with infections distributed across the farm area (Fig 4(A)). Therefore, modelling the individual status of every plant as a node would be impractical and computationally expensive, due to the varying contact points between plants, and the complex plant-pathogen-vector relationships as described in the previous section. Instead, the data may be aggregated to monitor the infection likelihood over larger areas of the farm (as depicted in Fig 4(B)). The wide dirt paths throughout a farm act as a soft barrier to BBTV spread, since most aphids are apterous (wingless) and are likely to move from leaf-to-leaf. Therefore, the subsections in a plantation may be described as nodes. Nodes that correspond to neighbouring subsections are linked, giving rise to an undirected network, as seen in Fig 3(A). Any subsection containing at least one infected plant may be considered 'infected' (Fig 4(C)).

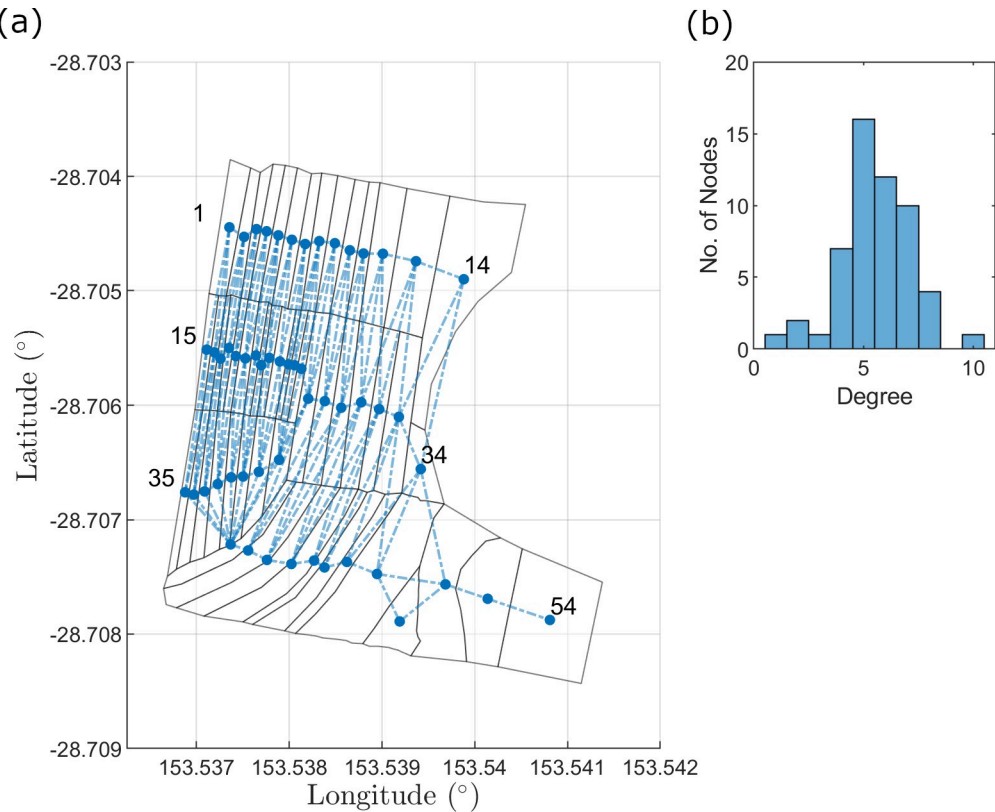

**Fig 3. Representative network structure of banana plantation.** (a) Labels indicate node index. Each plantation subsection is regarded as a node. Subsection coordinates are mapped using open source GRASS GIS software. (b) Degree-distribution of network. Highest-degree node: node 43 with 10 edges; Lowest-degree node: node 54 with 1 edge.

Furthermore, Dutta et al. [8] describe $M_s$ to propagate infection spread in the network through a single seed node. This is an unrealistic assumption for modelling BBTV spread in plantations, since it is possible for a plantation to have multiple latent infections upon first exposure to the virus. Therefore, $M_s$ must be adapted to accept multiple seed nodes. Additionally, since the scope of this paper is limited to the analysis of current trends and predictions to evaluate disease management strategies, we are not interested in inferring the initial seed node (s). Rather, the BBTV model considers the infected nodes observed in the first month of field data surveying to be the initial configuration of seed nodes at *t = 0* (see Fig 4(C)).

Finally, the SI model of Dutta et al. [8] is extended by explicitly considering the recovery of an infected node. A node is considered recovered if, upon inspection at time *t*, it contained at least one infected plant, while at time *(t+1)* no infected plants were found. Recall that if an infected plant is found within a certain subsection of a plantation, it is immediately rogued and removed. This may be considered as a recovery. Removing the infected plant(s) from its subsection reverts the corresponding node to a susceptible state, as the remaining plants within the subsection/node remain vulnerable to being infected, giving rise to a Susceptible-Infected-Susceptible (SIS) model.

## Parameters

The unique epidemiological characteristics of BBTV must be parameterised to fully capture the complex dynamics of BBTV transmission in plantations, and to extend on the current literature on BBTV.

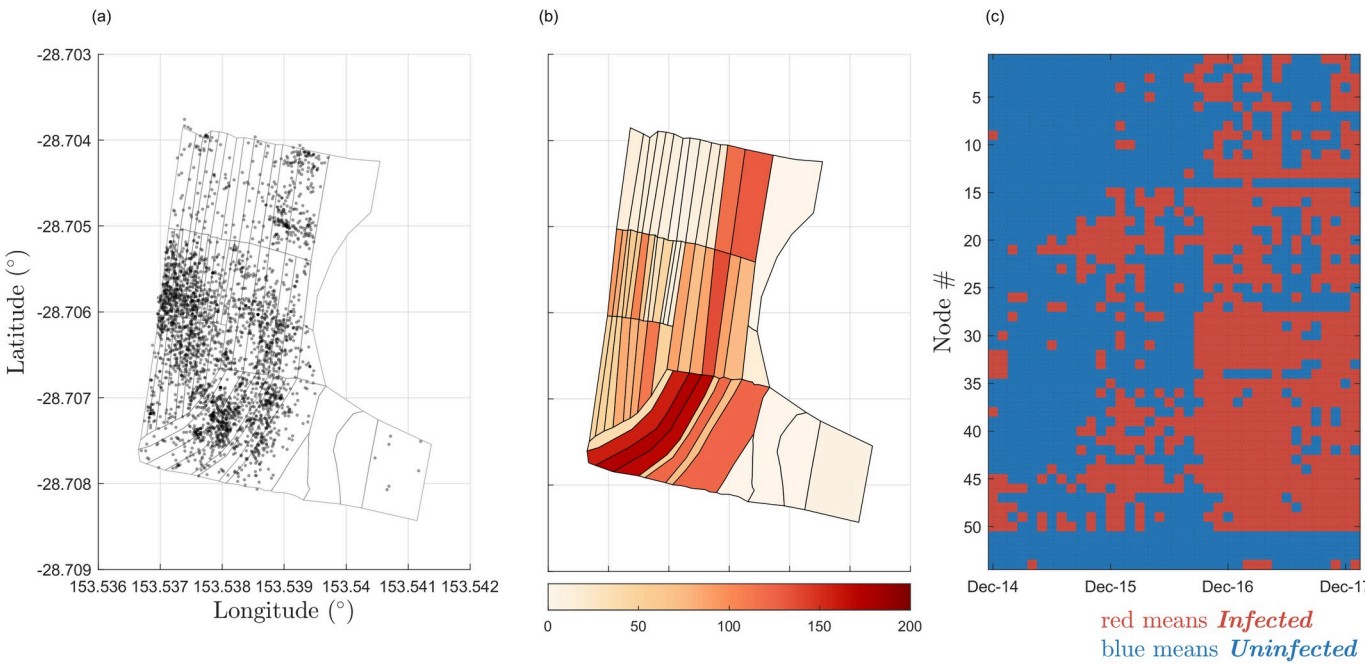

**Fig 4. Distribution of BBTV Infections.** (a) Distribution of BBTV infections over total observation period. Black points represent observed infections. (b) Discrete-space distribution of BBTV infections over total observation period. Infection coordinates are binned to plantation subsections. Subsections are coloured according to infection count. (c) Discrete spatio-temporal distribution of infection presence in plantation subsections across observation period. This is the final data used for parameter estimation. The first column of the data is used as the initial configuration for model simulation. The node # of the corresponding subsection detailed in Fig 3(A).

There are three parameters that are estimated in this model:

1. *Probability of recovery*, $\theta_0$: As infected plants are rogued and removed at monthly inspections of infected plantation visits, nodes may subsequently recover from an infected state to a susceptible state. Currently, there is no existing literature on BBTV recovery rates in field scenarios using current disease management strategies. Accurate estimates of the probability of node recovery in plantations could inform decision makers on the effectiveness of current roguing methods, inspection frequency and inspection accuracy.

2. *Neighbouring probability of infection*, $\theta_1$: The model $M_s$ created by Dutta et. al [8] proposes an infection probability for each of the neighbours of an infected node. This parameter is also relevant for our case study, as the aphid vector is likely to travel between neighbouring nodes. Allen [6] identifies that the probability of a BBTV infection is inversely proportional to the distance from a previously infected plant, as most aphid flights cover small distances.

3. *Non-neighbouring probability of infection*, $\theta_2$: While short distance flights are more likely to occur in banana plantations, long distance aphid vector transmission remains a possibility. This parameter operates on all infected nodes, whereby each node has a probability of infecting every non-neighbour. Aphids are also known to be restless and sensitive to small changes in the environment and have been shown to relocate to other plants due to overpopulation, harvesting activities and sudden changes in atmospheric weather conditions [9].

The operations of these parameters are summarised in Fig 5 below:

**Seasonality.** As highlighted by the infection counts presented in Fig 2(B), BBTV infectivity is influenced by seasonal changes in temperature, therefore it may be useful to identify changes in the posterior distribution of the parameters for different seasons. Allen [6]

identifies that detection efficiency, eradication efficiency and aphid activity are seasonally varying factors which greatly affect the spread of BBTV. Furthermore, Allen [10] confirms a seasonally varying leaf emergence rate in bananas, which informs detection and eradication efficiency. Anhalt and Almeida [11] observe temperature to be highly correlated with acquisition and inoculation efficiency, with peak transmission efficiency occurring at 25–30 degrees.

To accommodate for seasonal variation, we allow each parameter to be month dependent. While it may be theoretically possible to generate unique posterior estimates for each month, their accuracy may be greatly diminished by the lower amount of field data available for each monthly parameter. To maintain an effective sample size of observed data to inform each parameter while ensuring a clear differentiation between parameter counterparts, each parameter has been replicated by dividing and grouping months traditionally above or below the long-term average temperature. Months with an average temperature traditionally higher than the long-term annual average temperature, may be referred to as 'summer' months, and vice versa for 'winter' months. Summer months are given by September, October, November, December, January and February. Likewise, winter months are given by March, April, May, June, July and August.

Therefore, the final set of parameters are described by $\theta_{ij}$, with $i \in [0, 1, 2]$ indicating the parameter type (recovery, near, and long distance infectivity respectively) and $j \in [0, 1]$ indicating the season (summer and winter respectively).

The mechanisms of the forward-simulation model are summarised in Fig 6.

## Approximate Bayesian Computation (ABC)

The parameter $\phi = \{\theta_{00}, \theta_{10}, \theta_{20}, \theta_{01}, \theta_{11}, \theta_{21}\}$ may be inferred by its posterior density $p(\phi \mid y)$ given the observed dataset $y$. The posterior density can be written by Bayes' theorem as,

$$p(\phi \mid y) = \frac{\pi(\phi)\, p(y|\phi)}{m(y)} \tag{1}$$

where $\pi(\phi)$, $p(\phi \mid y)$ and $m(x) = \int \pi(\phi)\, p(y|\phi) d\phi$ are, correspondingly, the prior density on

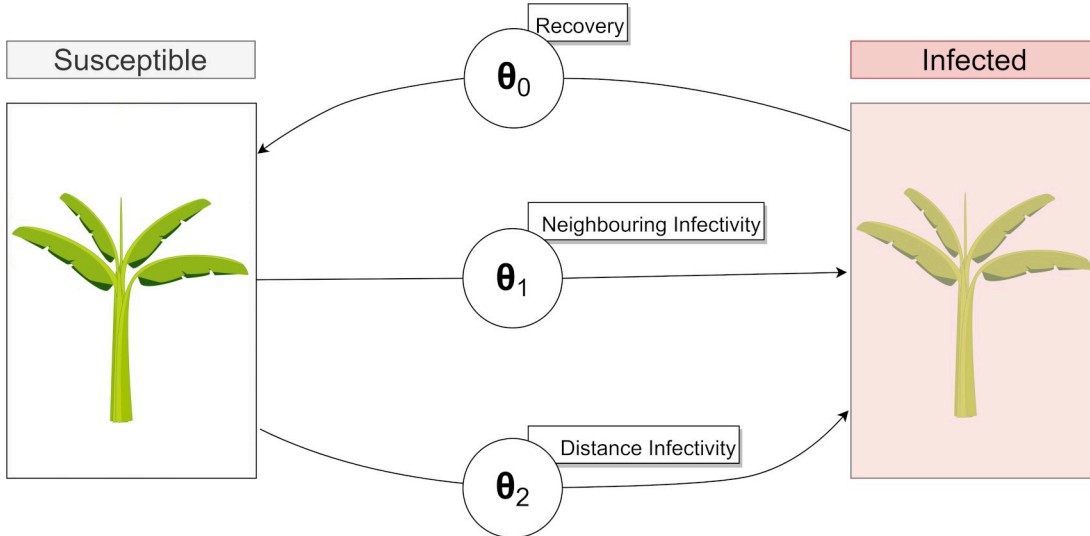

**Fig 5. State-flow diagram describing each parameter's influence on the state of a node.** A single Banana plant has been chosen to represent an entire subsection of bananas.

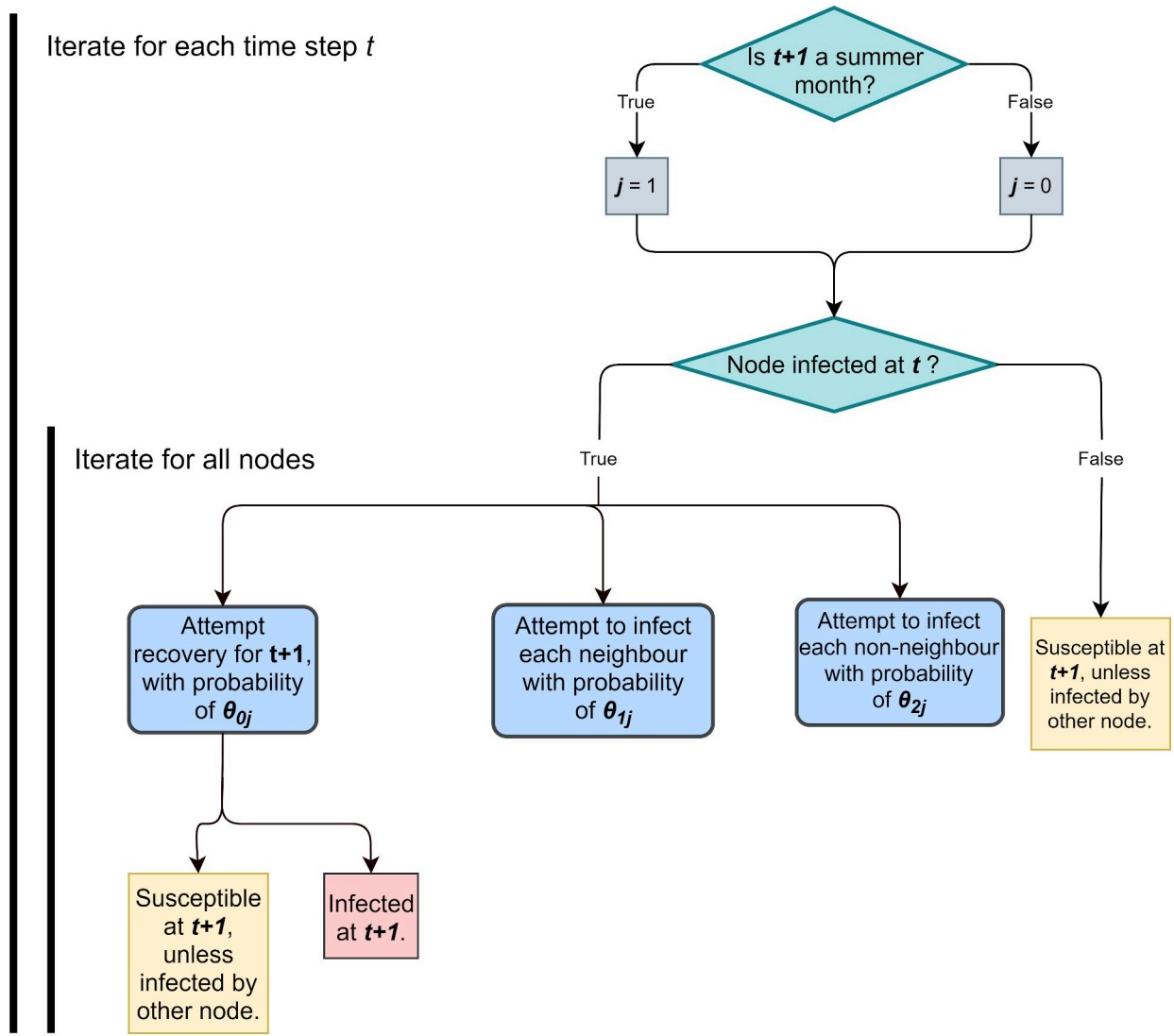

**Fig 6. Flow-chart describing BBTV model behaviour for each time step *t*.**

the parameter $\phi$, the likelihood function, and the marginal likelihood. The prior density $\pi(\phi)$ enables a way to leverage the learning of parameters from prior knowledge, such as the epidemiological characteristics and expert knowledge on BBTV [8].

Alternative modelling methods for describing the disease dynamics of vector-borne diseases within plantations have been previously covered in the epidemiological literature. In particular, the use of state-space epidemiological models with continuous-time processes have been demonstrated in a variety of scenarios, such as describing the disease spread of the Citrus tristeza virus (CTV) in a citrus orchard [12] and an aphid infestation in a sugar cane plantation [13]. Employing this class of models, with some simplifying assumptions on plantation shape and uniformity enables a relatively straightforward calculation of the likelihood function, $p(y \mid \phi)$, enabling the use of likelihood-based inference techniques such as Markov Chain Monte Carlo (MCMC) or Sequential Monte Carlo (SMC) to estimate the posterior distributions of parameters with high accuracy.

The use of a static, spatial network model to describe the spatiotemporal spread of a disease in a plantation presents a novel approach to this modelling problem, and joins a growing body of research in the application of epidemiological network models [14]. However, the use of network models to describe disease dynamics presents significant challenges for employing likelihood-based inference approaches due to the complex nature of network structures, resulting in an intractable likelihood function [8]. While MCMC has been successfully employed for some network modelling applications in contact networks, all applications required the assumption of a 'tree-like' contact network structure to resolve the likelihood function [8]. While these assumptions do not hold for the spatial network employed in our application, fortunately, simulating our network model is computationally cheap. This enables us to employ ABC, which provides the opportunity to sample from the approximate posterior density of the parameters [7].

ABC bypasses the evaluation of the likelihood function by instead simulating data from the model to generate an approximate posterior distribution. Due to the high dimensionality of the observed data, $y$, the data set is often reduced to a set of summary statistics, $S(y)$. Thus, ABC targets the posterior conditional on the summary statistics:

$$p(\phi|S(y)) \propto p(S(y)|\phi)\,\pi(\phi) \tag{2}$$

However, this too requires the evaluation of a typically intractable likelihood, $p(S(y) \mid \phi)$. Therefore, ABC approximates this intractable likelihood through the following integral:

$$p_\epsilon(S(y)|\phi) = \int_y K_\epsilon(\rho(S(x), S(y)))p(x|\phi)\,dx \tag{3}$$

where $\rho(S(x), S(y))$ is a discrepancy function that compares the simulated and observed summary statistics, and $K_\epsilon(\cdot)$ is a kernel weighting function with bandwidth $\epsilon$ that weights simulated summaries in accordance with their closeness to the observed summary statistic. The role of the discrepancy measure will become clear in the next section. While the integral in (3) is analytically intractable, it may be estimated by taking $n$ iid simulations from the model $\{x_i\}_{i=1}^{n} \sim p(x|\phi)$, evaluating their corresponding summary statistics $\{S_i\}_{i=1}^{n}$ where $S_i = S(x_i)$, and calculating the following ABC likelihood:

$$p_\epsilon(S(y)|\phi) \approx \frac{1}{n}\sum_{i=1}^{n} K_\epsilon(\rho(S_i, S(y))) \tag{4}$$

The unbiased likelihood estimator described in (4) is generally sufficient to obtain a Bayesian algorithm that targets the posterior distribution $p_\epsilon(\phi|S(y)) \propto p_\epsilon(S(y)|\phi)p(\phi)$. The summary statistics, $S(\cdot)$, discrepancy measure, $\rho(\cdot,\cdot)$, and tolerance value, $\epsilon$, utilised in the ABC method introduce approximation errors to the target posterior distribution. In order to minimise these errors, these factors must be chosen and tuned carefully to maximise accuracy while ensuring a computationally feasible operation [15].

**ABC algorithms.** The most basic implementation of ABC is known as rejection sampling [16]. In this algorithm, the parameter is estimated by generating model realisations $x$ corresponding to different parameter values $\phi$ promoted from the prior. The summaries $S(x)$ are computed and compared to $S(y)$ through the discrepancy measure $\rho(\cdot,\cdot)$. If the discrepancy between the simulated and observed summaries is lower than the tolerance, $\epsilon$, then the corresponding $\phi$ is accepted as part of the approximate posterior distribution.

The pseudo-code for an ABC rejection sampling scheme is provided below [17]:

```
for i ϵ 1: n do
  Draw φ ~ π(φ)
  Draw x ~ p(· | φ)
  Accept φ if ρ(S(x), S(y)) ≤ ε
end for
```

where $n$ is the number of iid samples to be taken from the prior $\pi(\phi)$.

In this paper, ABC is implemented through an MCMC algorithm to effectively estimate the posterior distributions of the recovery and spreading parameters of BBTV in the Newrybar banana plantation. ABC-MCMC [18] aims to improve the efficiency in comparison to ABC rejection sampling, by proposing parameter values locally around promising regions of the parameter space.

**Summary statistics.**   The summary statistics play an important role in the ability for ABC methods to effectively estimate the posterior distributions of parameters [15]. Summary statistics summarise observed or simulated data which can often be large, complex and high dimensional. Effective summary statistics characterise the influence of specific parameters on the model, so that varying parameter values result in observable changes in the reported summary statistics.

We find that the following summary statistics are informative about the model parameters:

1. $S_1$–A vector with entry $S_1(t)$ being the proportion of infected nodes at each time step $t$ (where $t = 1, \ldots, 38$).

2. $S_{10}$–A scalar computed as the total number of infected nodes summed over all time-steps $t$, that recovered by $t + 1$.

3. $S_{010}$–A scalar computed as the total number of susceptible nodes summed over all time-steps $t$, which became infected by $t + 1$. These nodes must not have an infected neighbour at time $t$.

4. $S_{011}$–A scalar computed as the total number of susceptible nodes summed over all time-steps $t$, which became infected by $t + 1$. These nodes must have at least one infected neighbour at time $t$.

The development of summary statistics to capture important features of simulated data is largely intuitive and requires some tuning and validation through simulation studies with dummy data. In developing our summary statistics, we consider the informativeness of each summary statistic to a specific parameter, while minimising dimensionality.

The summary statistics are informative as follows:

- $S_1$ describes the temporal characteristic of the simulated infection spread and is influenced by all three parameters.

- $S_{10}$ provides an indication of the recovery rate, thus corresponding to $\theta_0$.

- $S_{010}$ describes the number of infections occurring via a vector from a long distance, corresponding with $\theta_2$.

- $S_{011}$ describes the number of infections occurring via a vector from a neighbouring node, informing $\theta_1$.

Although there are several approaches available to weight summary statistics in ABC [19], this is not necessary for our application since the total variance and value of each summary statistic is sufficiently similar, such that each statistic has an equal influence on the discrepancy measure. This has been validated through a simulation study where dummy 'observed' data

has been simulated with known parameter values, and an ABC-MCMC algorithm with the above summary statistics is used to estimate these parameters.

Since the parameters are applied during different seasons, the summary statistics $S_{10}$, $S_{010}$ and $S_{011}$ have been replicated for summer (20 out of 38 months) and winter. Thus, we have a total of 7 summary statistics, one vector of length 38, and 6 scalars (3 for summer and 3 for winter).

It may be noted that while $S_1$ is a vector describing the infection rate for each time step $t$, all other summaries ($S_{10}$, $S_{011}$, $S_{010}$) are scalars–having been summed over all time steps $t$. While the use of a combined metric does reduce some of the information captured, we believe this is a reasonable approach for our application, especially given the great reduction in dimensionality to the final summary vector. Furthermore, we suggest that little information is lost in the aggregation. Consider the statistic $S_{10}$, for example. Under the assumed model, the number of nodes that recover from time $t$ to $t + 1$ is binomially distributed with the number of trials given by the number of infected nodes at time $t$ and "success" probability $\theta_{0j}$. Aggregating this statistic over the summer or winter months results in a sum of binomially distributed random variables each with a different number of trials and the same success probability. Using the properties of the binomial distribution, the sum is also binomially distributed. Thus, under the assumed model, the aggregated statistic carries the same information as the vector of statistics over the summer or winter months. A similar argument can be made for $S_{011}$ and $S_{010}$. This significantly reduces the length of the summary vector, enabling lower ABC tolerances and higher acceptance rates, and resulting in better approximations of the posterior distributions of the parameters [16].

**Discrepancy measure ($\rho(\cdot,\cdot)$).**   The Mean Squared Error (MSE) between the simulated and observed summary vector is utilised as a discrepancy measure for this case study. This may be described as follows:

$$\rho(S(x),\ S(y)) = \frac{1}{k}\sum_{j=1}^{k}(S_j(x) - S_j(y))^2 \tag{5}$$

where $S(x)$ is the summary vector of the simulated data, $S(y)$ is the summary vector of the observed data, and $j$ denotes an element of the summary vector, while $k$ represents the number of elements in each summary vector.

Here we incorporate all our summary statistics into a single discrepancy measure to compare observed and simulated data, as is standard in ABC analyses. An alternative approach may be to define a distance metric for each summary with its own tolerance (e.g. Ratmann et al. [20]).

**ABC-MCMC iterations.**   10 million (1e7) MCMC iterations were simulated to achieve the approximate posteriors for the model parameters in order to maintain a reasonable effective sample size. We opted for a burn-in of (1e5) MCMC iterations and thinned by a factor of 200 to reduce correlation between proposals along the MCMC chain. This results in approximately 40,000 unique samples after thinning and burn-in.

**ABC-MCMC tolerance ($\varepsilon$).**   Reducing the MCMC tolerance ($\varepsilon$) increases the accuracy of the posterior distribution, at the expense of computation time. An extensive simulation study was conducted which revealed that an MCMC tolerance of 23 was appropriate to provide accurate posteriors while ensuring a reasonable computation time.

**Priors.**   Given the novel nature and the lack of specific expert knowledge regarding these parameters, uniform priors with bounds of [0, 1] have been chosen for all parameters.

## Results and discussion

### Posterior distributions

As shown in Table 1, mean posterior recovery probability ($\theta_{0j}$) is influenced by the seasonal changes in temperature. While the mean posterior recovery probability for an infected node is 25.8% in summer ($\theta_{01}$), this increases to 30.67% for winter ($\theta_{00}$).

Mean posterior neighbouring infectivity ($\theta_{1j}$) is 2.12% higher in summer months compared to winter months (6.18% and 4.06% respectively), indicating minor seasonal dependence. Distant infectivity exhibits a similar dependence ($\theta_{2j}$), as the mean posterior probability during summer of 0.72% only decreases by 0.1% during winter months (Fig 7 and Table 1).

These results may be explained through a range of environmental factors. Higher posterior probabilities for node recovery in winter are likely due to lower inspection accuracy arising from lower leaf growth rates and decreased farming activity during this season. Research from the Department of Employment Economic Development and Innovation identify that bananas growing in the sub-tropical climates, such as the South-East Queensland area, are heavily influenced by temperatures: the rate of production is often significantly reduced in winter, sometimes to a rate of one leaf in 20 days [21]. In contrast, summer leaf emergence can be completed in around four days in tropical conditions. Since BBTV in banana plants is identified through observing visual symptoms of infections, identifying newly infected plants is much more likely during summer compared to winter, where a plant may be latently infected for months before displaying signs of infection. Therefore, the lower reported BBTV infection rates in winter would artificially increase the posterior probability of recovery in this season.

Higher neighbouring and distant infectivity in summer is likely due to more weather events in this season, and inoculum acquisition sensitivity to tropical temperatures. According to the historical monthly averages of climate data provided by the Bureau of Meteorology, South-East Queensland experiences significantly higher wind speeds during summer at an average maximum gust of 131.7 km/h compared to 93.7 km/h during winter [22]. Similarly, average rainfall during summer is 113.7 mm for 11 days compared to 84.2 mm for nine days during winter [22]. A higher frequency and intensity of such weather events are likely to perturb the aphid vector, as observed by Claflin et al. [9], increasing the risk of neighbouring and long-distance transmission. Additionally, Anhalt & Almeida [11] identify that *P. nigronervosa* provides peak inoculum acquisition and transmission efficiency between 25–30 degrees Celsius, which is historically experienced during the summer months in South-East Queensland. Higher seasonal rates of inoculum transmission, in conjunction with increased vector activity during this period could account for greater counts of neighbouring and long-distance infections during the summer.

The estimated pairwise posterior distributions of model parameters highlight key operative characteristics of the model. As seen in Fig 8, the estimated univariate posterior distribution for the posterior probability of recovery is largely symmetric, while the posterior probability of neighbouring and distant infectivity ($\theta_{1j}$) remains positively skewed.

Negative correlation is evident between neighbouring infectivity ($\theta_{1j}$) and distant infectivity in the corresponding months ($\theta_{2j}$), which indicates that different combinations of these parameters can generate similar summary statistics. This is an intuitive correlation since the

**Table 1. Mean posteriors of parameters.**

|  | Summer ($j = 0$) | Winter ($j = 1$) | Δ |
|---|---|---|---|
| Recovery ($\theta_{0j}$) | 25.8% | 30.67% | +4.87% |
| Neighbouring infectivity ($\theta_{1j}$) | 6.18% | 4.06% | -2.12% |
| Distant infectivity ($\theta_{2j}$) | 0.72% | 0.62% | -0.1% |

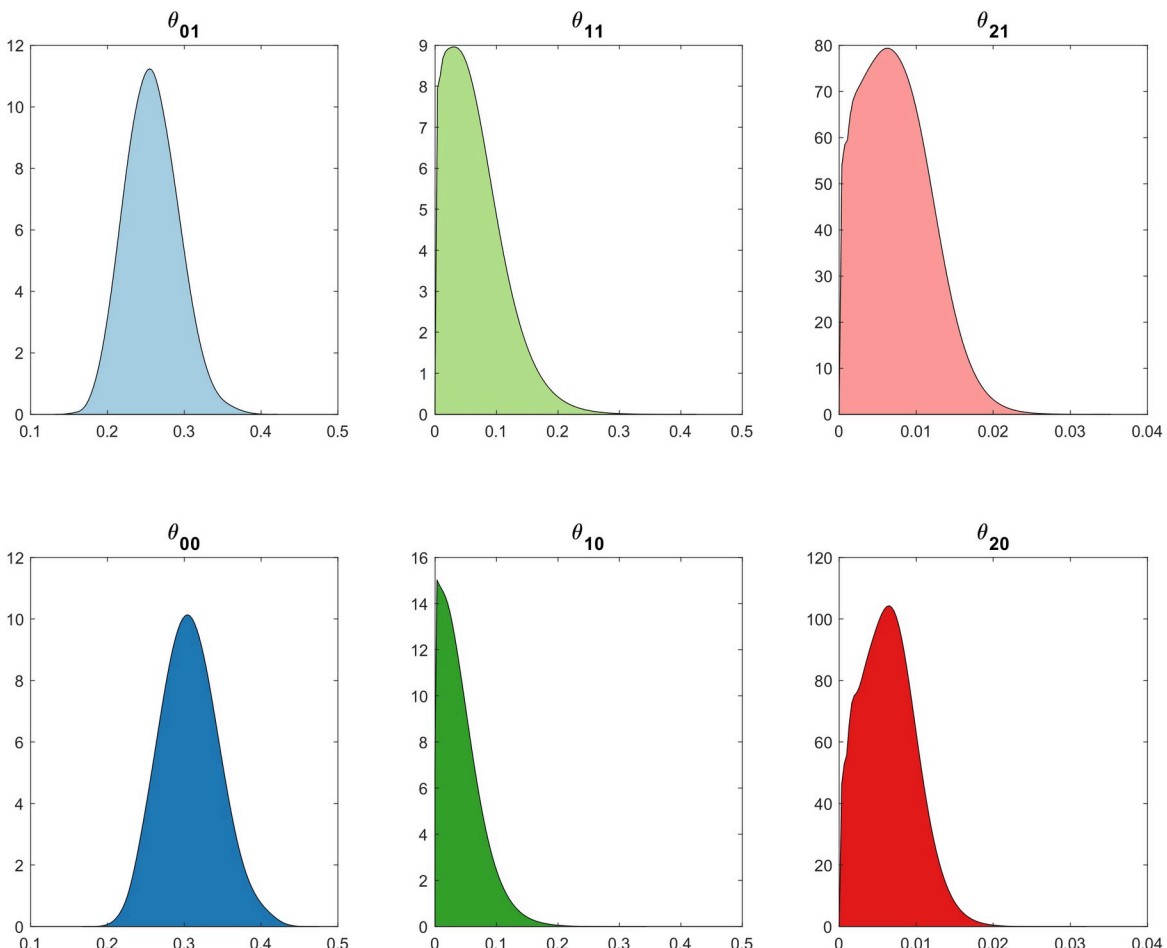

**Fig 7. Approximate posterior distributions of parameters.** Posterior densities are coloured to correspond with their respective seasonal counterpart, with darker colours representing winter seasons.

total number of infected nodes in each month is a sum of the number of nodes infected by a neighbour or over long-distance. Therefore, if a high probability of neighbouring infectivity is proposed in a model simulation, a low probability of distant infectivity will be more likely to result in the observed summary statistics.

## Posterior forecasting

Our modelling framework allows us to provide a posterior infectivity forecast for all nodes over a 6-month period (Fig 9). The forecast is generated by running the BBTV forward simulation model with parameters obtained from a random sample from the posterior distribution.

Each node begins with a 100% or 0% infection probability in month 38, since this is the last known time-step provided to the model as the initial configuration for each of the three simulation scenarios. The subsequent forecasted probabilities of infection for each node converge to a steady state probability of 45%. The forecasted infection probabilities for nodes infected in the last known time-step are characterised by a depreciation in infection probability, and an increasing variance in individual infection probability at each subsequent month. In contrast, forecasted probabilities for initially uninfected nodes begin with a high variance in the group, progressively decreasing in subsequent months.

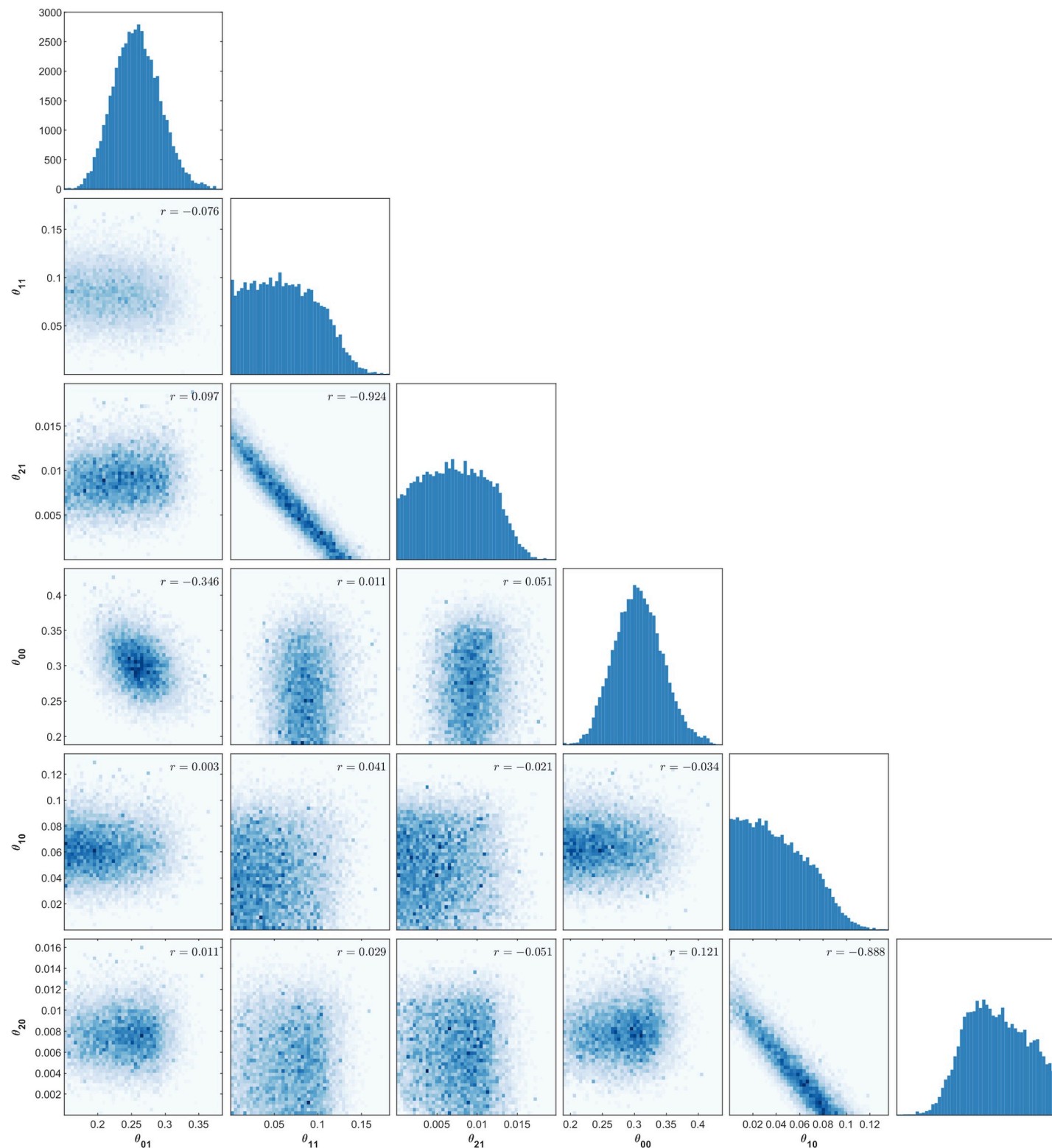

**Fig 8. Estimated univariate and pairwise posterior distributions using ABC-MCMC.**

Fig 9(B) provides a posterior forecast of node infectivity which applies the posterior counterparts for summer months to all months ($\theta_{i1}$). Compared to Fig 9(A), this results in a higher steady state probability of 57%. Furthermore, while previously uninfected nodes in both figures exhibit a steep increase in forecasted node infectivity by the first month, the subsequent infection probabilities flatten to the steady state infection probability in Fig 9(A) and continue to increase in Fig 9(B). This is likely due to the higher neighbour infectivity and distant infectivity probabilities during summer.

This may be confirmed through Fig 9(C), which provides a 6-month posterior forecast of node infectivity, utilising the posterior counterparts of winter months to all months ($\theta_{i0}$). Unlike Fig 9(B) and 9(C) displays a gradual increase in forecasted probability of node infectivity over subsequent months, tending to a lower steady-state node infectivity probability of 40%.

**Discrete-space posterior probability forecast.**    Fig 10(A) and 10(B) visualise the 1-month posterior probability forecasts for infection in each node. Fig 10(A) depicts the posterior infection probabilities for infected nodes in the last observed time-step, while Fig 10(B) depicts the posterior infection probabilities for previously uninfected nodes.

Posterior forecasts from the BBTV model indicate that previously infected nodes are likely to remain infected, consistently reporting an infection probability of approximately 74% for all nodes. Since the forecasted month (month 39) occurs during the summer period, the posterior distributions of the summer counterparts ($\theta_{i1}$) are utilised for this simulation.

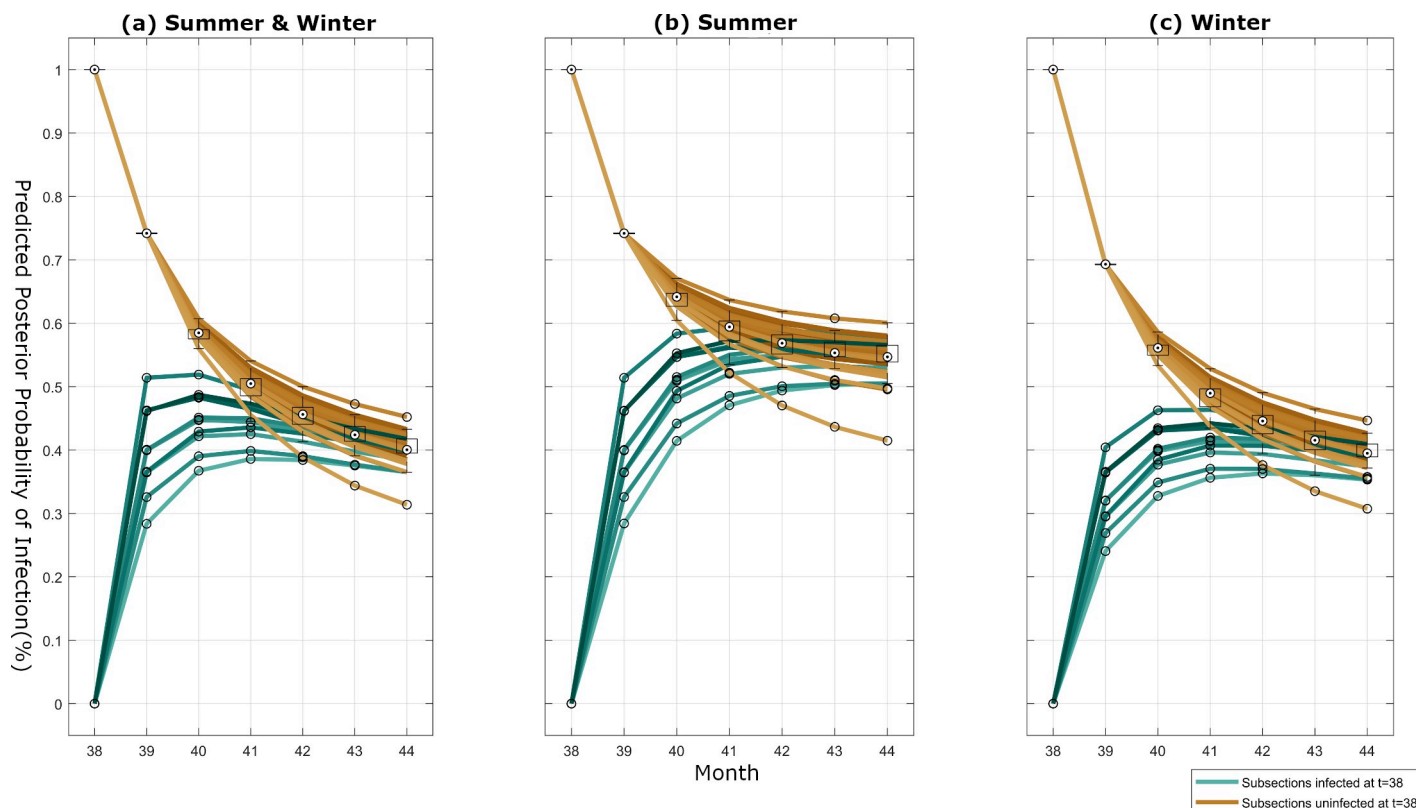

**Fig 9. 6-month posterior forecast of node infectivity.** (a) Simulated using both summer and winter posterior counterparts. (b) Simulated using only summer posterior counterparts. (c) Simulated using only winter posterior counterparts. Green lines indicate subsections infected at the last known timestep $t = 38$, while brown lines indicate subsections in a susceptible state at $t = 38$.

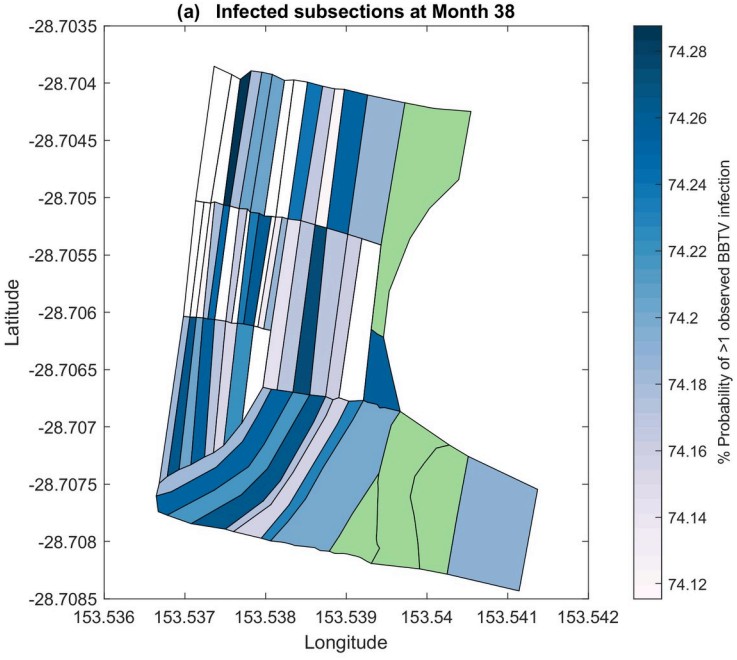

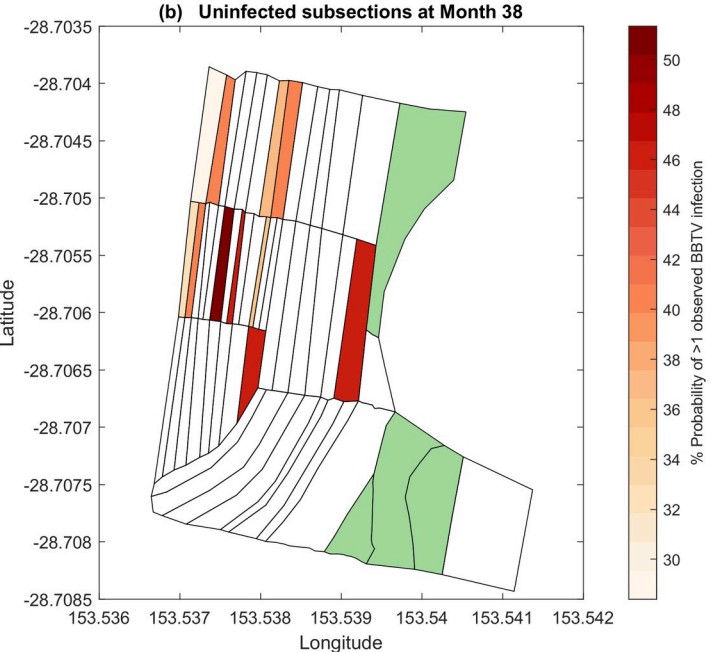

**Fig 10. 1-month discrete-space posterior forecast of node infectivity.** (a) Posterior probability of node infection for previously infected nodes. (b) Posterior probability of node infection for previously uninfected nodes. Subsections highlighted in green indicate nodes not planted with bananas.

Fig 10(A) and 10(B) indicate that the highest posterior predicted infection probability is associated to nodes with a high number of neighbours. This is likely due to the greater

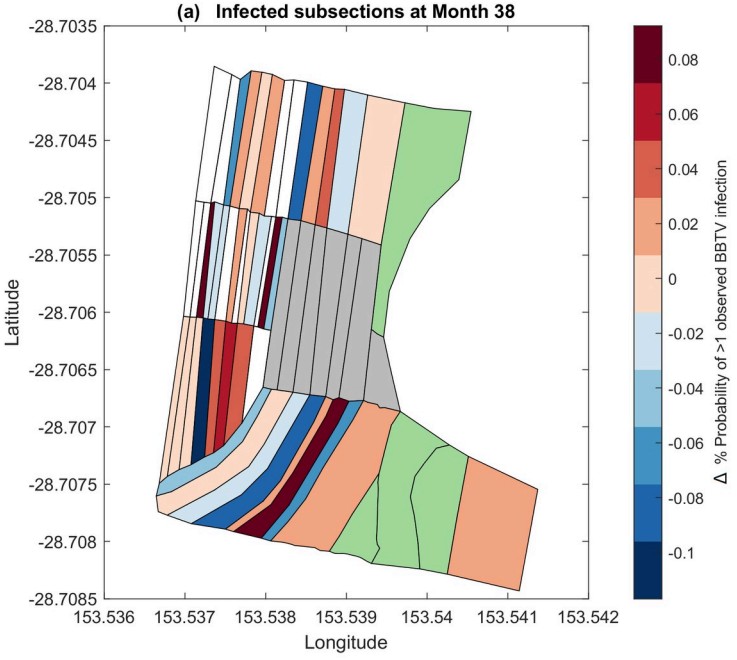

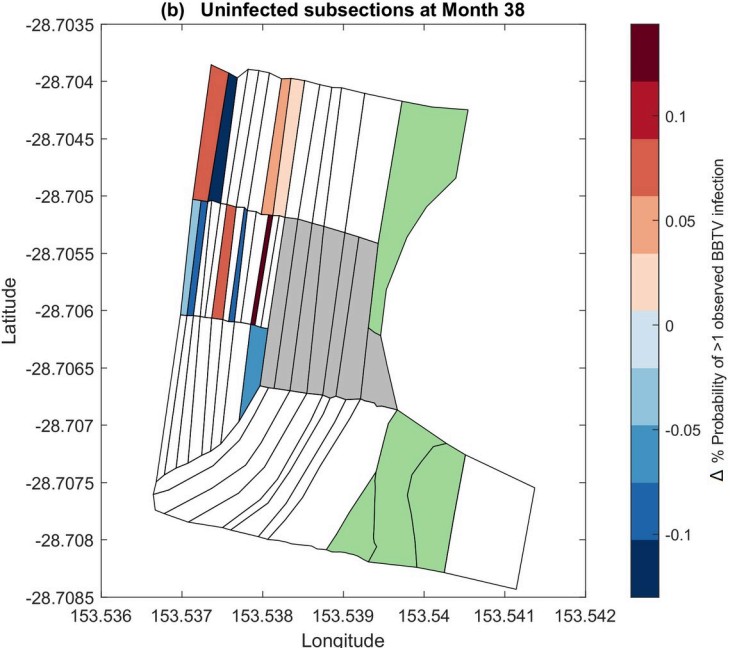

**Fig 11. 1-month discrete-space posterior forecast of node infectivity.** (a) Posterior probability of node infection for previously infected nodes. (b) Posterior probability of node infection for previously uninfected nodes. Subsections highlighted green indicate nodes not planted with bananas. Subsections highlighted grey indicate nodes that have been 'cleared' according to the simulation.

application of neighbouring infectivity ($\theta_{1j}$) on this node, as a high number of neighbours increases the probability of the aphid vector to infect this node.

**Alternate plantation organisation.** The model may be utilised to explore the reconfiguration of a banana plantation, to observe the effect of clearing a subsection to mitigate the local spread of BBTV. The clearing of a subsection may be considered equivalent to freezing its corresponding node in a susceptible state for all time steps *t*. Through this method, any long- or short-range vector transmission to a cleared node will remain unsuccessful.

Field surveyors with the National Banana Bunchy Top Project expressed interest in exploring the impact of clearing the central section of the farm (nodes 20 to 34). Fig 11(A) and 11(B) describe the 1-month posterior infection probability forecasts for month 39 if nodes 20 to 34 (in grey) were cleared.

When compared to the forecasts from Fig 10(A) and 10(B), removing nodes 24 to 32 results in some nodes have a higher posterior infection probability by up to 0.8%, while others decrease by a maximum of 1%. When averaged across all remaining nodes in the network, there is an insignificant reduction in the forecasted posterior infection probabilities. Further configurations of cleared areas may be explored upon recommendations by stakeholders.

## Posterior predictive checking

In addition to the 38 months of field data provided by the BBTV Prevention Program from December 2014 to January 2018 which was used to estimate the model parameters (training dataset), an additional 7 months of field data till August 2018 was provided, which may be utilised as validation dataset. In order to conduct posterior predictive checking, the model was set to simulate the posterior infection probabilities of each node in month ($t + 1$), given an initial configuration of the infected nodes at month *t*. The validation data for month ($t + 1$) may then be compared to the corresponding posterior infection probabilities to identify model accuracy.

**Binomial deviance loss.** The accuracy of the posterior predicted infection probabilities for a subsection may be identified through its deviance from the validation data. This may be calculated through the binomial deviance loss function, which provides a discrepancy between a prediction and its corresponding binomial validation data (true node infection state). The loss function is defined as follows:

$$yP - \log(1 + e^P) \tag{6}$$

where *y* is the true node infection state in month ($t + 1$), and *P* is the log odds of the corresponding posterior prediction. A deviance loss of 0 indicates a completely correct prediction of the true infection state of a node at time ($t + 1$), while a loss of -2 indicates that the predicted state of a node at time ($t + 1$) is totally deviant from the true infection state recorded in the validation dataset.

Fig 12 displays the binomial deviance loss for each prediction for each subsection, coloured according to the validation dataset used. Prediction accuracy is largely unaffected by the validation set used, with the mean deviance loss across all subsections being -0.6; a 40% improvement compared to a random prediction. Posterior predictions for certain subsections are consistently excellent (see nodes 27–32, 43–50), due to consistent incidence levels in these areas of the farm.

**Receiver-Operating Characteristic (ROC) curve.** A ROC Curve illustrates the diagnostic ability of a binary classifier system as its discrimination threshold is varied. The ROC curve is created by plotting the true positive rate (TPR) against the false positive rate (FPR) at various threshold settings. Fig 13 displays the ROC curve of the model, which demonstrates an Area-Under-the-Curve (AUC) of 0.65, generally considered to be a "fair" model. This metric is a

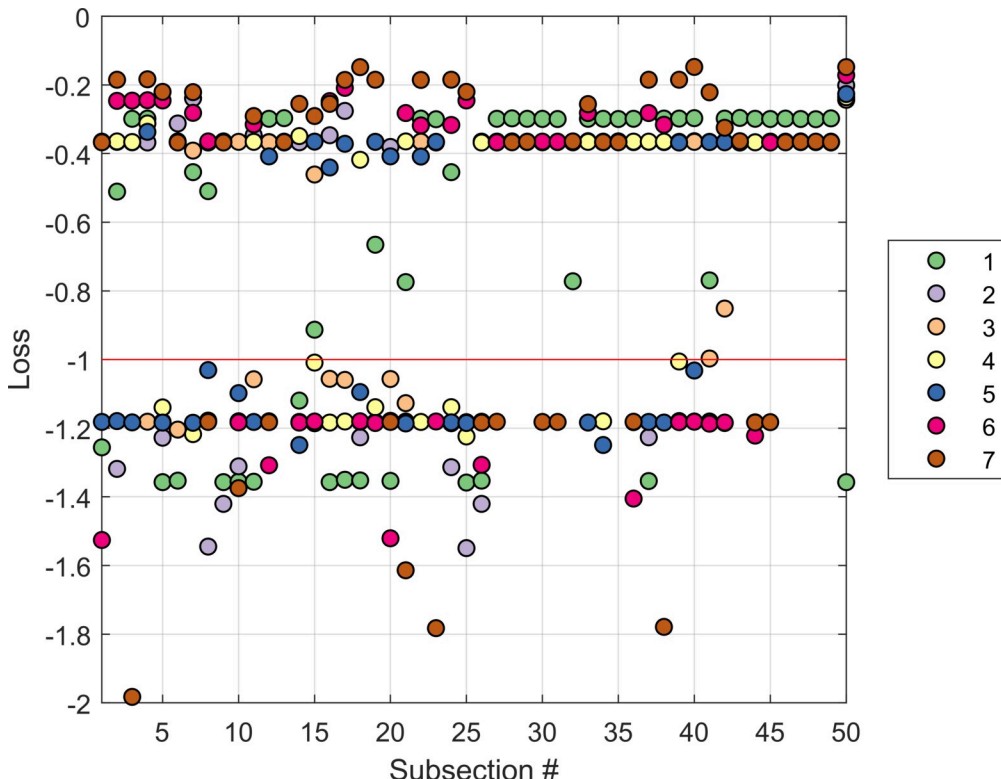

**Fig 12. Binomial deviance loss aggregated for all predicted months in a 6-month period.** Each dot represents the binomial deviance for a prediction corresponding to a subsection, coloured according to the predicted validation data set. The red line represents the deviance loss of a posterior prediction of 50% (random).

general indicator of the posterior prediction confidence, as a larger area under the curve would indicate higher true positive rates at lower threshold levels. The ROC curve is well above the diagonal line, indicating that it performs markedly better than random predictions, particularly at higher thresholds.

## Implications and future research

Monitoring the nodal recovery ($\theta_{0j}$), near infectivity ($\theta_{1j}$) and distant infectivity ($\theta_{2j}$) rates over time would serve to metricise the effectiveness of disease management strategies. For example, a long-term reduction in $\theta_{0j}$ could be a signal of increased aphid resistance to chemicals used in current pesticide routines, or of an increase in latent infections across the plantation. Correspondingly, a long-term increase in $\theta_{1j}$ and $\theta_{2j}$ could point to higher aphid activity across the plantation, potentially indicating a reduction in inspection accuracy.

Providing site inspectors with specific high-risk areas through posterior predictive modelling, could improve inspection accuracy. Furthermore, the identification of 3 or 4 high-risk areas in a plantation opens the opportunity of increasing the site inspection frequency from monthly to fortnightly inspections, while limiting the inspection coverage to these high-risk areas.

The model has several limitations. Firstly, it does not account for the geographical characteristics of the plantation. The steepness and height of an area of the plantation would affect the ability for the aphid vector to travel to neighbouring subsections/nodes. Furthermore, the area represented by each node, and the distance to neighbouring nodes is quite varied and

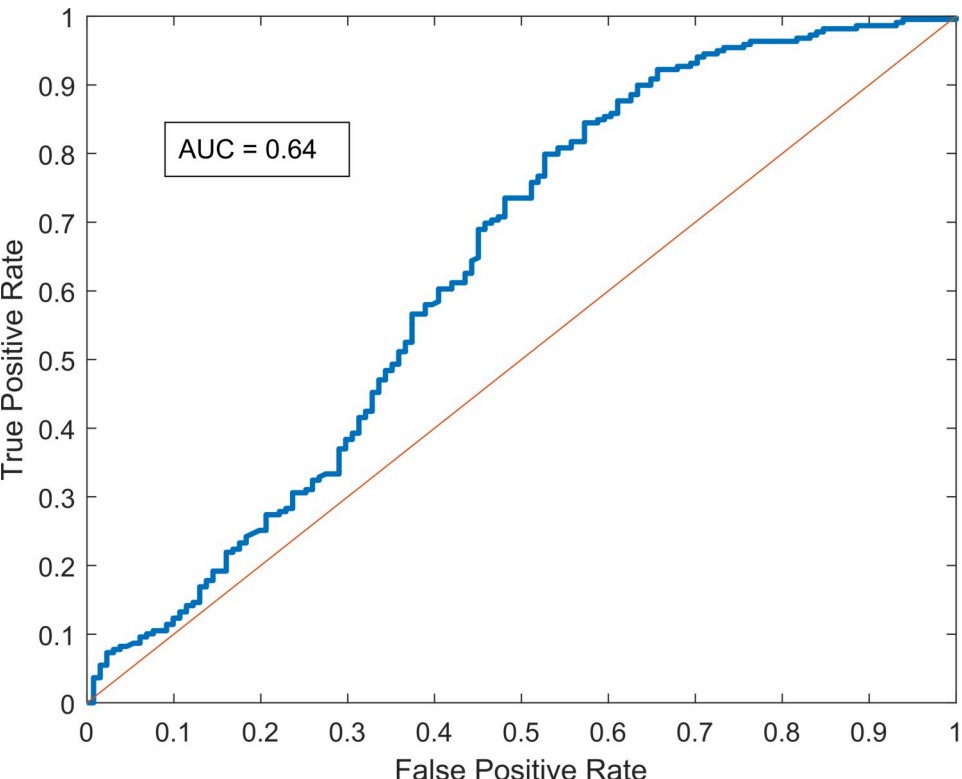

**Fig 13. Receiver-Operating Characteristic (ROC) curve for the BBTV forward-simulating model.** Predictions across all validation data sets have been aggregated to construct this curve.

would be likely to influence the posterior probability of neighbouring and distant infectivity due to the greater distance for the aphid vector to travel, and the number of potential host plants in a subsection. Secondly, environmental factors such as the wind speed and direction, and extreme weather events, are not considered in this model. Higher wind speeds and extreme weather events would be more likely to perturb and relocate the aphid vector, resulting in greater infectivity rates across the plantation. Thirdly, the methodology places limits on the achievable resolution of posteriors, since the model relies on the presence of clear divided areas in a banana plantation to identify nodes and establish a network. Furthermore, since the collection of field data relies on visual observation, the accuracy of model parameters largely relies on the inspection accuracy which is influenced by seasonality and environmental factors, in addition to inspector experience and fatigue. Lastly, the model only considers a subsection to be infected if at least one infection has been reported in that subsection, whereas the field data collected at Newrybar would enable the calculation of infection counts in each subsection/node, as well as the total number of infected leaves present in a plant, and thus in a subsection/node. These factors would significantly improve model accuracy and informativeness. Extending the model to address these limitations should be considered for future research.

## Conclusion

This paper has adapted and extended upon current network-based disease models implemented in an ABC framework. A forward-simulating network-based SIS model has been created which simulates the spread of BBTV across the subsections of a banana plantation, by parametrising nodal recovery, neighbouring infectivity and distant infectivity across summer

and winter. Findings from posterior results achieved through ABC-MCMC indicate seasonality in all parameters, which are influenced by correlated changes in inspection accuracy, temperatures and aphid activity. This model enables the simulation, monitoring and forecasting of various disease management strategies, which may support policy-level decision making and inspector experience. Introducing higher dimensional field and weather data will improve model accuracy and utility; an area to be explored for future research.

## Supporting information

**S1 Zip. Source code.** MATLAB code for network model, summary statistics, and ABC-MCMC parameter estimation algorithm.
(ZIP)

**S2 Zip. Source data.** Raw.CSV data of infections recorded from banana plantation from Dec-2014 to Jan-2018.
(ZIP)

**S1 Document. Supporting information MCMC convergence analysis.** Document providing convergence analysis of MCMC algorithm, algorithm choice and sensitivity analysis of ABC-MCMC tolerance parameter.
(DOCX)

**S1 Fig. Post burn-in trace plots of ABC-MCMC chain for all parameters.** Referred to in *S1 Document*.
(TIFF)

**S2 Fig. Post burn-in (1e6) and thinned (factor of 200) lag-*k* autocorrelation plots of ABC-MCMC for all parameters.** Referred to in *S1 Document*.
(TIFF)

**S3 Fig. Approximate posterior means across multiple tolerance thresholds.** Referred to in *S1 Document*.
(TIFF)

## Acknowledgments

The authors would like to acknowledge the support of Hort Innovation in providing the data required for this project, and Barry Sullivan, Dr. John Thomas and Dr. Kathy Crew for their expert advice on BBTV throughout this project.

## Author Contributions

**Conceptualization:** Abhishek Varghese, Antonietta Mira, Kerrie Mengersen.

**Data curation:** Abhishek Varghese.

**Formal analysis:** Abhishek Varghese, Antonietta Mira.

**Funding acquisition:** Kerrie Mengersen.

**Investigation:** Abhishek Varghese, Christopher Drovandi, Kerrie Mengersen.

**Methodology:** Abhishek Varghese, Christopher Drovandi, Antonietta Mira, Kerrie Mengersen.

**Project administration:** Abhishek Varghese.

**Resources:** Kerrie Mengersen.

**Software:** Kerrie Mengersen.

**Supervision:** Christopher Drovandi, Kerrie Mengersen.

**Validation:** Abhishek Varghese, Christopher Drovandi, Antonietta Mira.

**Visualization:** Abhishek Varghese.

**Writing – original draft:** Abhishek Varghese.

**Writing – review & editing:** Abhishek Varghese, Christopher Drovandi, Antonietta Mira, Kerrie Mengersen.

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
