## [Decision Letter · Decision Letter 0]

14 Oct 2019

Dear Dr Varghese,

Thank you very much for submitting your manuscript 'Estimating a novel stochastic model for within-field disease dynamics of banana bunchy top virus via approximate Bayesian computation' for review by PLOS Computational Biology. Your manuscript has been fully evaluated by the PLOS Computational Biology editorial team and in this case also by independent peer reviewers. The reviewers appreciated the attention to an important problem, but raised some substantial concerns about the manuscript as it currently stands, mostly regarding the justification and details of the particular methodology used to address the problem at hand. While your manuscript cannot be accepted in its present form, we are willing to consider a revised version in which the issues raised by the reviewers have been adequately addressed. We cannot, of course, promise publication at that time.

Sincerely,

Konstantin B. Blyuss

Guest Editor

PLOS Computational Biology

Virginia Pitzer

Deputy Editor

PLOS Computational Biology

[LINK]

Reviewer's Responses to Questions

**Comments to the Authors:**

Reviewer #1: Referee report on: PCOMPBIOL-D-19-01505

It is not clear what the network is. How many nodes, how many links, what is the degree distribution? Is this a fully spatial network, apart from the long-range transmission? Do patches separated by dirt tracks define nodes? Are all adjacent patches considered as neighbours? Would be useful to see some summary statistics of this spatial network.

Data on which the parameter estimation has been done is never presented, plotted or analysed. I would expect that at least some of the paper presents and analyses this data.

Why are there no comparisons between the true data and model output? Would like to see how output from the models with the best parameters compares to the true data.

The simulation seem to be a discrete time simulation with an arbitrary update windows, so how do you compare abstract time to real time that it is present in the real data? Is the abstract time-step set to some real time like day or week or so on? It is never mentioned over what time window has the plantation been observed and some plots of how the number of infected patches evolved in time.

See my comment about the lack of analysis of the data on which the inference is based on.

At some point the authors talk about SI model but their model is SIS.

What is the initial condition for the simulation?

Nodes are sometimes referred to as patches then subsections and other versions, can there be some consistency?

Supplementary material in my opinion is more important than some of the figures in the paper. These essentially try to make a case that the model is good/valid. I would move this to the main part of the paper and would make a better case that the model is valid by comparing model output to existing data over the whole period of collection. Or by devising a test whereby one divides the existing data in “used for inference” and “used for validation” and tests performed over different such divisions to validate the model properly.

So the role of the vector is simply the long-range infection?

I am not sure if the choice of statistics is the best for comparing data to simulation.

First, I would like to see the data. Is it simply, time, patch, status? Is there seasonality in the collected data?

For reproducibility purposes is the code available on Github? What about the data?

Line 114: over a range of plantation

Line 132-133: what do the authors mean by “the new BBTV

model does not estimate for an initial seed node”? Do they mean that their model is not able to infer the initial seed?

Line 191: I think it should be /pi(\\phi) not \\phi(x) in the integral

Line 315: In the sentence “Fig 6 provides a posterior infectivity forecast for all subsections over a 6-month period” what does “subsections” refer to? Is this each node/patch?

Line 363-364: “if subsections 20 to 34 (in grey) were cleared” – is this equivalent o removing nodes?

Line 408-410: again this is not an SI model, in epidemiology this is an SIS model.

Line 468-472: what ranges of values can formula 2 take and what value means good fit versus bad fit?

Reviewer #2: Review of “Estimating a novel stochastic model for within-field disease dynamics of banana

bunchy top virus via approximate Bayesian computation” by Varghese et al. (PCOMPBIOL-D-19-01505)

This paper describes the development and fitting of a stochastic SIS model of banana bunchy top virus (BBTV) to some novel data from a plantation in Australia. The model is fitted to the available data using ABC-MCMC. The underlying disease dynamics involve an intermediate aphid vector, however data on aphid populations are unknown, and instead the authors simplify the underlying model to look at aggregate probabilities of localised transmission and long-range transmission between “nodes” in a network, where each node corresponds to a group of plants in spatially contiguous areas. This approach has the advantage of not having to model the aphid vector dynamics, at the cost of modelling at quite a course spatial level. They extend their model to use different transmission and recovery probabilities in different seasons (summer/winter) in order to try to assess the impact of weather / temperature etc. on the transmission dynamics.

The results suggest some marginal impact of season on localised transmission potential, and negligible impact of season on long-range transmission and recovery.

Overall I think the paper looks fine. It’s a relatively straightforward application of ABC methodology to a novel data set, and would be of interest to the readers of the journal call. The authors state a “significant” impact of season on localised transmission potential, but I think that the approximate posteriors overlap a lot here and so I might argue that the effect is not that “significant”. In my opinion there also needs to be more discussion of the ABC tolerance used in the ABC-MCMC, and what impact that might have made on the results as well. Perhaps subtle differences are being masked by the approximations introduced in the ABC, and this warrants further discussion. Another discussion point could be whether the use of a combined metric might be averaging out some of the information in the data,.

Minor remarks:

Abstract (and elsewhere): I think that technically the authors are using an SIS model, not an SI model per se, because there are transitions back to the S class (and hence why they observe endemic equilibriums in their forecasts).

L57, L94: Sometimes there are capital letters in e.g. “North-Eastern” (L94) but other times lower-case letters e.g. “south-east” (L57). These should be consistent throughout the manuscript.

L128: Sentence beginning, “Secondly, ….” does not read well. Suggest a re-write to clarify what the authors mean.

L144: “rogueing” should be “roguing”.

L160: The figure citations in the text are all out by one (and throughout the rest of the manuscript). This line should read “Fig. 1” and not “Fig. 2”, and this carries through the paper.

Equation 4: Should this have a 1/n outside the sum?

L242: Some more exposition about how the authors figured out which summary measures were informative would be interesting. What criteria did they use?

L245: It might be worth clarifying in the notation that S1 is specified for each time point (all the other summary measures are explicitly stated to be at time point t, except the first).

P13: How many replicates simulations were used to generate the estimate in equation 4? This should be stated somewhere.

Fig 4: It looks like the plot in the bottom-right has had its axes flipped compared to the other plots. Is this correct?

P21: Why did the authors choose to remove the grey squares? Why these and not any other squares? Possibly Fig 8 might be more informative if the differences were mapped, rather than the absolute values.

L384: “...their priori...” doesn’t make sense.

L411: Here the authors use “MCMC-ABC” but elsewhere they use “ABC-MCMC”.

Reviewer #3: See attachment

**Have all data underlying the figures and results presented in the manuscript been provided?**

Reviewer #1: No: no data seen or even plotted

Reviewer #2: Yes

Reviewer #3: Yes

PLOS authors have the option to publish the peer review history of their article (what does this mean?). If published, this will include your full peer review and any attached files.

Reviewer #1: No

Reviewer #2: No

Reviewer #3: No

---

## [Decision Letter · Decision Letter 1]

10 Mar 2020

Dear Mr. Varghese,

Thank you very much for submitting your manuscript "Estimating a novel stochastic model for within-field disease dynamics of banana bunchy top virus via approximate Bayesian computation" for consideration at PLOS Computational Biology. As with all papers reviewed by the journal, your manuscript was reviewed by members of the editorial board and by several independent reviewers. Reviewers 1 and 3 have suggested accepting your manuscript for publication as is. Reviewer 2 has mentioned a few additional very minor issues that they would like to ask you to sort out, and once this is done, they would not need to review the paper again, and the paper can then be accepted for publication.

Sincerely,

Konstantin B. Blyuss

Guest Editor

PLOS Computational Biology

Virginia Pitzer

Deputy Editor

PLOS Computational Biology

[LINK]

Reviewer's Responses to Questions

**Comments to the Authors:**

Reviewer #1: I thank the authors for addressing all my comments.

I am happy to recommend the paper for publication.

Reviewer #2: Review of “Estimating a novel stochastic model for within-field disease dynamics of banana

bunchy top virus via approximate Bayesian computation” by Varghese et al. (PCOMPBIOL-D-19-01505)

This is a review of a revised version of the above manuscript. I thank the authors for taking the time to consider and act on the comments from the first review. In my opinion, the manuscript is now much improved and clearer as to the intent, novelty and limitations of the study. As such I think it is suitable for publication, subject to a few minor amends as highlighted below.

One point is that there are now a lot of figures. I appreciate that the authors have moved some figures from the Supplementary Information to the main paper in response to some comments from the other reviewers. However, I think the balance has shifted a bit too far. For example, Fig. 7 is effectively the diagonals of Fig. 8, and as such I would remove Fig. 7 and just reference Fig. 8. I would also put all the MCMC convergence plots into the Supp Mat (e.g. Figs 9 and 10). Also, Fig 11 is a sensitivity analysis type plot and I think it should be in Supp Mat also.

There is also no Fig. 12 as far as I can see, so again, please take care with figure numbering.

Minor remarks:

Abstract, L33: I don’t think the “predict for” is correct here. Would suggest something like:

“Our models can be used to quantify the effects of …. BBTV spread, and can be used to predict high-risk areas in this plantation.”

L131: Should probably reference the exact GIS software used.

L281: Should probably be \\rho(S(x), S(y)) to be consistent with previous paragraph applying the distance metric to summary measures.

L297-300: Please make it clear that these are summed across all time points. This is not highlighted in the text until the following page, and I think it would help clarity to make this explicit here.

Table 1: Should the summer and winter be j = 0 and j = 1 respectively (not j = 1 and j = 2)?

L431: This is interesting, but I disagree slightly with the author’s interpretation. To my mind, what they have done is conduct a sensitivity analysis around what would happen to the posterior marginal Cis if they were to change the tolerance. They suggest that since the CIs don’t change much, then the “chosen tolerance value is reasonable for our application”. I might suggest that what they’ve done is assess the relative impacts of small changes, rather than the absolute impacts. I would suggest rewording this to something along the lines of, “small changes to the tolerance do not seem to impact on the approximate posterior uncertainties here, and so we decide to keep with our initial choice of tolerance.”

This also relates to my earlier question (apologies for not making this clear in the first review), that a choice of an aggregated discrepancy measure might average out some information in the data (an alternative is to use multiple distance metrics around each summary measure, each with a different tolerances e.g. Ratmann et al. 2009).

L592: “Significant” seasonality?

Reviewer #3: Whilst I am not fully convinced that the network model considered is more complex than the spatial models I referenced in my original review, I do believe that this work makes a nice contribution to applying ABC to epidemic models. I have no further comments to raise.

**Have all data underlying the figures and results presented in the manuscript been provided?**

Reviewer #1: Yes

Reviewer #2: None

Reviewer #3: Yes

PLOS authors have the option to publish the peer review history of their article (what does this mean?). If published, this will include your full peer review and any attached files.

Reviewer #1: No

Reviewer #2: No

Reviewer #3: No
---

## [Editor Report · Decision Letter 2]

15 Apr 2020

Dear Mr. Varghese,

We are pleased to inform you that your manuscript 'Estimating a novel stochastic model for within-field disease dynamics of banana bunchy top virus via approximate Bayesian computation' has been provisionally accepted for publication in PLOS Computational Biology.

Best regards,

Konstantin B. Blyuss

Guest Editor

PLOS Computational Biology

Virginia Pitzer

Deputy Editor

PLOS Computational Biology

---

## [Editor Report · Acceptance letter]

12 May 2020

PCOMPBIOL-D-19-01505R2 

Estimating a novel stochastic model for within-field disease dynamics of banana bunchy top virus via approximate Bayesian computation

Dear Dr Varghese,

I am pleased to inform you that your manuscript has been formally accepted for publication in PLOS Computational Biology. Your manuscript is now with our production department and you will be notified of the publication date in due course.

With kind regards,

Laura Mallard
